# RNA N^6^-Methyladenosine Affects Copper-Induced Oxidative Stress Response in *Arabidopsis thaliana*

**DOI:** 10.3390/ncrna10010008

**Published:** 2024-01-19

**Authors:** Bishwas Sharma, Ganesan Govindan, Yongfang Li, Ramanjulu Sunkar, Brian D. Gregory

**Affiliations:** 1Department of Biology, University of Pennsylvania, Philadelphia, PA 19104, USA; sharmab@sas.upenn.edu; 2Department of Biochemistry and Molecular Biology, Oklahoma State University, Stillwater, OK 74078, USA; ganeshmssrf@gmail.com (G.G.); yongfangli@gmail.com (Y.L.); 3Department of Genetic Engineering, SRM Institute of Science and Technology, Kattankulathur 603 203, Tamil Nadu, India

**Keywords:** epitranscriptome, long non-coding RNAs, plant stress response, RNA methylation, ribosome occupancy, transcript stability

## Abstract

Recently, post-transcriptional regulation of mRNA mediated by N^6^-methyladenosine (m^6^A) has been found to have profound effects on transcriptome regulation during plant responses to various abiotic stresses. However, whether this RNA modification can affect an oxidative stress response in plants has not been studied. To assess the role of m^6^A modifications during copper-induced oxidative stress responses, m^6^A-IP-seq was performed in Arabidopsis seedlings exposed to high levels of copper sulfate. This analysis revealed large-scale shifts in this modification on the transcripts most relevant for oxidative stress. This altered epitranscriptomic mark is known to influence transcript abundance and translation; therefore we scrutinized these possibilities. We found an increased abundance of copper-enriched m^6^A-containing transcripts. Similarly, we also found increased ribosome occupancy of copper-enriched m^6^A-containing transcripts, specifically those encoding proteins involved with stress responses relevant to oxidative stressors. Furthermore, the significance of the m^6^A epitranscriptome on plant oxidative stress tolerance was uncovered by assessing germination and seedling development of the *mta* (N^6^-methyladenosine RNA methyltransferase A mutant complemented with *ABI3*:*MTA*) mutant exposed to high copper treatment. These analyses suggested hypersensitivity of the *mta* mutant compared to the wild-type plants in response to copper-induced oxidative stress. Overall, our findings suggest an important role for m^6^A in the oxidative stress response of Arabidopsis.

## 1. Introduction

All living organisms rely on their ability to respond to the environmental changes around them for their growth and survival. In general, plant species are sessile in nature, and therefore are constantly subjected to unfavorable abiotic and biotic environmental conditions. As a result, plants have evolved a variety of complex cellular response pathways that help them respond to both abiotic and biotic stresses, ultimately restoring and maintaining cellular homeostasis [1,2,3]. These important response pathways are usually activated by biochemical molecules generated in the cells. Reactive oxygen species (ROS) are one such key species of biomolecules that are naturally formed as a byproduct of metabolism and, if left unchecked, can cause cellular and DNA damage [4,5]. Therefore, the levels of ROS generated due to routine metabolic processes are usually regulated by ROS scavenging proteins that sequester them and prevent them from building up in the cells, protecting them from damage [1,4,5]. However, almost all stressors including drought [6], pathogens [7], unfavorable temperature [8], salinity, and heavy metals [9] in the soil can induce a higher than usual amount of ROS in plants and induce cellular damage inhibiting growth and development—a condition known as oxidative stress. During oxidative stress, many ROS also simultaneously work as signaling molecules to help plants respond to this stress by inducing signaling pathways that activate genes key to their survival [10].

More recently, ROS molecules have been shown to affect epigenetic pathways of gene regulation in Arabidopsis, rice, and other plant species [6,11]. In Arabidopsis, histone deacetylases 19 and 9 (HDA19 and HDA9) were found to be inactivated as a result of oxidative stress, potentially leading to increased histone acetylation and altered gene expression of the associated loci [12]. Additionally, in pokeweed (*Phylotolacca americana* L.), the frequency of DNA methylation was shown to be partly mediated by ROS [13]. These studies suggest that ROS-mediated stress response affects gene expression at different levels. Since ROS are relatively short lived, the ROS response pathways are likely to also be regulated at the post-transcriptional level to allow plants to regulate their response quickly and precisely. While studies have investigated the signal transduction of genetic and epigenetic pathways involving ROS, no study has yet investigated the possible epitranscriptomic regulation of oxidative stress response.

In the last decade, the diversity and functions of epitranscriptomic modifications of messenger RNA (mRNA) have been studied in many organisms including more recently in plant species such as Arabidopsis, rice, and maize [14,15,16]. To date, hundreds of RNA modifications have been characterized [17] but functional studies of RNA modifications have been limited to a few modifications due to lack of molecular technology. Of these modifications, N6-methyladenosine or m^6^A is the most prevalent mRNA modification found in most eukaryotic species. Functional studies have shown m^6^A to be associated with mRNA stability [18,19], localization [20], splicing [21], and translation [22,23,24]. Recent studies have identified the role of m^6^A in stress response. In plants, the loss of function mutant of the m^6^A writer protein METHYLTRANSFERASE A (MTA), *mta* mutant, results in developmental delay as well as improper response to salt stress and cold stress [18,24,25]. Similarly, another study showed that ectopic expression of human m^6^A demethylase FTO in rice and potatoes makes them more tolerant to drought [26]. Since the oxidative stress response pathway is intricately linked with many important abiotic and biotic stresses plants endure, it is highly likely that m^6^A is also implicated in this pathway. To date, no plant studies have investigated the role of epitranscriptomic marks specifically under oxidative stress. Furthermore, in addition to mRNA, many studies have shown that non-coding RNAs are m^6^A methylated [27,28] but to date no plant study has highlighted the possible regulation of non-coding RNA upon stress.

In this study, we investigate the change in the transcriptome-wide m^6^A methylation profile during Arabidopsis oxidative stress response. There are several ways to induce oxidative stress in plants. For instance, studies have shown that exposure to high levels of copper ions (Cu^2+^) induces excess levels of ROS accumulation in plants [10,29,30,31], and that this can provide a controllable system to induce oxidative stress in Arabidopsis. Using excess Cu^2+^ treatment, we performed m^6^A RNA immunoprecipitation sequencing (me-RIP-seq) to investigate changes in the m^6^A status of the transcripts. Additionally, we also looked at changes in the polysome binding of the transcripts whose m^6^A enrichment status were most responsive to copper-induced oxidative stress. Furthermore, we examined the impact of m^6^A enrichment on the long non-coding RNAs identified in our study.

## 2. Results

### 2.1. Reduction in Global m^6^A Levels Due to Loss of MTA Leads to Lack of Response to Copper-Induced Oxidative Stress

To determine the effect(s) of the epitranscriptomic mark m^6^A on plant oxidative stress response, we first investigated whether global reduction of methylation in the Arabidopsis transcriptome would have any effect on the ability of plants to respond to this stress. To do so, we used the previously well-characterized post-embryonic loss-of-function mutant of the Arabidopsis m^6^A writer protein MTA (*mta ABI3*:*MTA*; referred to hereafter as *mta*) [14]. Since complete loss of MTA is embryonically lethal, several studies have used this mutant that expresses the MTA writer protein only in the embryonic stages of development by using the embryo-specific ABI3 promoter. Thus, we compared the survival and growth of this *mta* mutant as compared to wild-type Col-0 seedlings grown on control MS agar plates and MS plates supplemented with copper sulfate at increasing concentrations of 100 and 125 μM. We found that both Col-0 and *mta* displayed overall decreased seedling growth and survival in the presence of increasing concentrations of copper. However, it was obvious that the *mta* mutant seedlings demonstrated a significant hypersensitivity to high levels of copper as compared to Col-0 plants. Relatedly, we also observed that, when grown vertically in the presence of increasing copper sulfate concentrations, the *mta* mutant displayed severely stunted primary root growth and seedling development compared to Col-0 plants (Figure 1a). Taken together, these data suggest that m^6^A methylation of Arabidopsis transcripts is critical to the plant’s ability to properly respond to oxidative stress induced by higher concentrations of copper.

### 2.2. Copper-Induced Stress Increases the Abundance of Transcripts Encoding Proteins Involved in Responses to Oxidative Stress and Pathogen Defense

To understand the overall effect of oxidative stress on the Arabidopsis transcriptome, we subjected 20-day-old seedlings to oxidative stress by spraying them with Cu^2+^ solution (see Methods for description) and measured the change in transcript abundance using high-throughput polyA^+^ selected RNA sequencing (mRNA-seq) after 24 h of treatment. Differential expression (DE) analysis identified a total of 79 transcripts that displayed a statistically significant (adjusted *p*-value < 0.05) change in their overall abundance in the treated samples relative to the untreated samples, with most of the messenger RNAs (mRNAs) displaying an increase in abundance (Figure 1b). Overall, although the transcriptome-wide RNA change was modest, Gene Ontology (GO) analysis revealed that transcripts associated with highly relevant biological processes were significantly altered in their abundance levels (Figure 1c). Among these were transcripts encoding proteins that were previously shown to play significant roles in oxidative stress responses, such as members of the glutathione transferase family (*GSTU2*, *GSTU4*, *GSTU9*, *GSTU19*, and *GSTU24*) as well as a cysteine-rich receptor-like kinase (*CRK11*) [32,33]. As expected, copper-induced transcripts are associated with GO terms such as the oxidation-reduction process, response to oxidative stress, and response to cadmium ion (Figure 1c).

### 2.3. Thousands of m^6^A Peaks Are Specifically Induced or Depleted upon Copper-Induced Oxidative Stress in mRNA

Next, we profiled the dynamic changes in m^6^A methylation in the overall Arabidopsis transcriptome upon oxidative stress treatment. To do this, we carried out m^6^A-IP-seq, in which RNA fragments containing m^6^A are isolated using an m^6^A-specific antibody (SYSY:202008), and these immunoprecipitated sequences are subjected to high-throughput RNA sequencing as previously described [34]. Using the peak calling algorithm MACS2, we identified m^6^A peak regions in RNAs both in the presence and absence of copper stress. In total 17,772 and 17,247 high-confidence m^6^A peaks were shared between the three biological replicates of copper-treated and control samples, respectively (Figure 2a). By comparing these peaks between the stress and control groups, we found a notable difference in the m^6^A profile of the transcriptome with and without copper stress. Altogether, we identified 2578 m^6^A peaks that were identified only under stress conditions and 3105 that were found only under control conditions. These dynamic m^6^A peaks corresponded to the 2316 and 2579 transcripts, respectively, which were subsequently referred to as stress-enriched and control-enriched m^6^A-containing transcripts. Furthermore, approximately 14,671 m^6^A peaks that did not change upon stress were referred to as common or shared transcripts between the two conditions. We performed GO analysis to investigate what functions the proteins encoded by these three groups of m^6^A methylated transcripts may play and found that many of the transcripts that contain the stress-enriched m^6^A peaks are associated with the terms oxidation-reduction process, innate immune response, protein import into nucleus, and anthocyanin production (Figure 2b). Conversely, the group of transcripts containing control-enriched m^6^A peaks were related to negative regulation of cell death, flowering, and other developmental processes (Figure 2b). These results suggest that transcripts that help plants respond to oxidative stress were gaining m^6^A methylation specifically in response to stress treatment.

Since m^6^A-IP-seq does not provide nucleotide level resolution and gives a collection of mostly 150–200 base pair (bp) peak regions, we detected the enriched motifs in these m^6^A peak regions to provide a better idea of the actual methylated sites. Using the motif recognition algorithm HOMER, we found a previously annotated and potentially plant-specific m^6^A motif “UGUA” as the top hit in these m^6^A peak regions from both control and copper stress m^6^A-IP-seq datasets (Figure 2c) [24,35]. These results strongly suggest the high quality of our m^6^A-IP-seq datasets.

### 2.4. m^6^A Tends to Be Localized in mRNA 3′ UTRs and Its Localization Pattern Is Sensitive to Copper-Induced Oxidative Stress

To visualize the localization of m^6^A along the lengths of mRNA transcripts, we calculated the density of m^6^A-IP-seq reads in 100 nucleotide (nt) bins representing the 5′ UTR, coding region (CDS), and 3′ UTR regions of all detected mRNA transcripts. We observed that overall, both the control and copper-treated samples displayed the same 3′ UTR bias in accumulation of reads in the m^6^A-IP-seq as compared to the background polyA^+^ RNA-seq datasets, which shows a strong enrichment of sequencing reads in the CDS region (Figure 2d). This 3′ UTR localization pattern of m^6^A is well established across animal and plant species. Since we observed a large overlap between overall m^6^A peaks between stress and control RNA, we next wanted to investigate whether there might be a change in m^6^A localization pattern due to stress. To do this, we focused on the group of transcripts whose methylation status was sensitive to stress treatment. For this group of transcripts, we calculated the methylation frequency of each nucleotide position around the start and stop codon for the stress-enriched or control-enriched m^6^A-containing transcripts and plotted it against the many transcripts that do not change methylation status (labelled shared) upon stress treatment (Figure 3a). Interestingly, we found that the frequency of methylation in the 5′ UTR was higher for transcripts containing stress-enriched m^6^A compared to the other groups. Similarly, the percentage of unique m^6^A peaks that overlapped with the 5′ UTR and start codons were also higher in transcripts that were either enriched or depleted in m^6^A upon stress treatment compared to transcripts that showed no change in m^6^A (Figure 3b). Overall, these results suggest that m^6^A peaks that are sensitive to oxidative stress treatment have a higher probability of being deposited at the 5′ UTR or CDS regions, upstream of where m^6^A is primarily found in the 3′ UTR. The mechanism by which this change in position is dependent on the enrichment/depletion of m^6^A warrants further studies.

### 2.5. Transcripts That Contain Copper Stress-Enriched m^6^A Are More Likely to Show Increased mRNA Levels

It has been previously suggested that m^6^A can affect transcript stability and abundance. A recent study from our group showed that transcripts that are enriched for m^6^A upon cold stress showed a positive trend in transcript abundance [24]. To test for any such correlations that exist in the context of oxidative stress, we used m^6^A profiling data and the RNA-seq data. After determining the fold change in expression for each transcript, we categorized the values based on whether the transcript contains m^6^A that is affected by copper treatment and plotted them on a box and whisker plot (Figure 3c). Overall, we found that transcripts with copper stress-induced m^6^A peaks displayed a significant increase in transcript abundance compared to those transcripts that were copper-depleted m^6^A or unchanged m^6^A peaks. The group of transcripts that lost their m^6^A methylation status upon copper treatment showed, on average, a more negative change in transcript abundance in the context of copper-induced oxidative stress response (Figure 3c). However, it should be noted that approximately ~18% (12 out 69 transcripts) of significantly increased transcripts in our mRNA-seq data were found to acquire stress-induced m^6^A upon copper stress compared to none of the decreased mRNA transcripts (9 in total) that showed induced m^6^A upon copper stress. Furthermore, among transcripts containing copper-enriched m^6^A, 27 showed a >50% decrease, whereas 139 displayed a >50% increase in mRNA levels after copper stress treatment, and these observations warrant further investigation. In general, this study suggests that gaining m^6^A during copper-induced oxidative stress tended to correlate with increased abundance for the transcripts. A few examples showing such correlations between the mRNA-seq and m^6^A-IP-seq in the presence of copper-induced oxidative stress are presented in browser views (Appendix A).

### 2.6. Transcripts with Copper Stress-Induced m^6^A Are More Likely to Be Polysome Associated

While stress-induced m^6^A was associated with a change in overall transcript abundance, the actual number of differentially expressed transcripts upon copper stress treatment in our study was still relatively low, as discussed previously. We therefore investigated whether m^6^A had other downstream effects on the transcript fate, specifically on translation. Although not a perfect measure of translation, we looked at polysome association as a proxy to study the effect of m^6^A on the ratio of transcripts that were bound to polysomes. To do this, we generated polysomal sequencing libraries by pulling down polysome fractions and extracting the RNA bound to them. Using polysome occupancy as a measurement, we plotted the change in occupancy of transcripts under stress to measure the effects on translation due to m^6^A (Figure 3d). From this analysis, we observed a relatively higher and statistically significant polysome association of transcripts containing stress-enriched m^6^A compared to those containing control-enriched or shared m^6^A. To further investigate which transcripts are responsible for the higher average polysome association among those that were copper-enriched, we broke down the category into those that show positive value for polysome occupancy change and those that show a negative value. Gene Ontology analysis revealed that transcripts relevant to oxidative stress response terms such as “response to oxygen containing compounds” and “defense related terms” showed increased polysome association in the copper-enriched m^6^A-containing transcripts (Figure 3e). These transcripts that are enriched for m^6^A upon stress showing increased binding to polysomes are candidates for investigating the molecular mechanisms of m^6^A regulation of translation in the context of oxidative stress in the future.

### 2.7. Long Non-Coding RNAs (lncRNAs) and m^6^A Association

Various classes of non-coding RNAs (ncRNAs) have previously been shown to contain m^6^A in many plant studies that performed global profiling of m^6^A [27,28]. Although the most highly abundant ncRNAs—ribosomal RNAs (rRNAs) and transfer RNAs (tRNAs)—are non-polyadenylated and therefore not captured by our m^6^A-IP-seq experiments, many other types of ncRNAs including long non-coding RNAs (lncRNAs) as well as precursors for microRNAs (miRNAs) and snoRNAs have been found to contain a polyA^+^ tail. A recent study conducted a global profiling of lncRNAs in Arabidopsis [36] using both polyA^+^ and polyA^−^ methods and found that a significantly higher number of lncRNAs (1352) were polyadenylated as opposed to those that were not (198), suggesting that lncRNAs were almost seven times more likely to be polyadenylated than not. Therefore, our polyA^+^ RNA-seq data can be used for studying m^6^A profiles in polyadenylated ncRNAs under oxidative stress. Using the annotations for different classes of ncRNAs loci from the ENSEMBL database, we compared the m^6^A peak information to detected m^6^A peaks found within the ncRNA regions. Among all methylated ncRNAs in our dataset, lncRNAs had the highest number of transcripts followed by general ncRNAs that were not categorized into the other types (Figure 4a). Furthermore, these methylated transcripts, when separated by the effect of oxidative stress on the m^6^A methylation, show that there are various lncRNAs and ncRNAs that are enriched for m^6^A upon copper treatment. Next, we inquired whether the ncRNA transcripts containing stress-induced m^6^A showed a similar pattern of positive RNA abundance change upon stress as protein-coding mRNAs of this category. Surprisingly, these ncRNA transcripts instead showed an overall negative transcript abundance upon stress, contrary to the overall trend of stress-induced m^6^A-containing mRNA transcripts (Figure 4b). This contradictory pattern suggests that m^6^A-mediated RNA response might be different between coding and non-coding RNA, particularly mRNAs as compared to lncRNAs. Further exploring this observation is an interesting question for future research.

## 3. Discussion

Oxidative stress is a complex physiological phenomenon that occurs simultaneously with, or a result of, almost every abiotic and biotic stress condition faced by plants [1,2,3,4,5]. It is crucial for plants to respond to oxidative stress to protect their cellular components from damage that can result from accumulation of ROS [4,5]. As such, it is highly likely that the regulation of gene expression in the oxidative stress response pathway happens at genetic, epigenetic, and possibly epitranscriptomic levels. While such regulation in the genetic and epigenetic levels is relatively well studied, the role of epitranscriptomic RNA regulation in response to oxidative stress is largely unknown. Our study suggests that epitranscriptomic RNA modification m^6^A is also involved in oxidative stress responses in plants.

Exposure of plants to excess copper ion is known to induce oxidative stress, triggering molecular response pathways that help them cope with the physiological changes brought about by the treatment [10,29,30,31]. Although our treatment did not trigger a massive shift in the RNA expression profile, the transcripts that showed statistically significant differential expression during copper treatment were mostly the genes that are known to be responsive to oxidative stress treatment, such as those belonging to the glutathione transferase family as well as other relevant functions.

In contrast to the modest number of transcripts showing a statistically significant change, we saw many transcripts that contained m^6^A which was either enriched or depleted upon oxidative stress. Overall, a significantly more positive change in transcript abundance was seen in transcripts with stress-enriched m^6^A. This suggests that enriched m^6^A is positively associated with transcript abundance. On the other hand, transcripts that showed no change in m^6^A levels under stress generally did not show changes in transcript abundance. While the exact mechanism of how copper stress-enriched m^6^A may lead to increased abundance of certain transcripts has not been fully established, it is possible that m^6^A reader proteins (YTH domain-containing proteins) that bind to this modification are involved in this process. Recent studies have suggested the YTH domain-containing proteins ECT2/3/4 are involved in transcript stabilization of transcripts associated with ABA stress responses [37]. Future experiments involving the mutants of these reader proteins under oxidative stress would be valuable to address this possibility.

The m^6^A peak distribution across mRNA features as well as the normalized m^6^A frequency plot shows that a small fraction of these stress-enriched or depleted m^6^A peaks are slightly more likely to be found on the 5′ UTR, start codon, and CDS regions than the peaks whose presence does not change upon stress. However, a mechanism that explains why these “upstream” peaks could affect the transcript stability differently is yet to be understood. On the other hand, it has been shown in mammalian systems that a 5′ UTR m6A can affect translation via an m6A reader protein where it can increase the rate of translation of methylated transcripts via a cap-independent translation mechanism [23]. We were curious to see if there was any effect on the translation status of methylated transcripts specifically under stress conditions. Our results show that overall polysome occupancy, which we are considering a proxy metric for translation status, is changing more positively for the transcripts containing stress-enriched m^6^A. A breakdown of this group of transcripts shows that oxidative stress response and defense response genes are driving the average polysome occupancy change to be more positive. Thus, the average of both metrics of gene expression (transcript abundance and polysome association upon oxidative stress) displayed a positive increase for those copper stress-enriched m^6^A-containing transcripts (Figure 5). It is, however, important to note that a considerable number of the transcripts in these m^6^A categories show the opposite/negative trend in transcript abundance and polysome association. Therefore, there is likely an unknown mechanism that provides a much-needed functional specificity to m^6^A’s role, one that most likely includes a m^6^A reader-dependent mechanism—reading of these changes by a specific m^6^A reader protein.

Non-coding RNAs play essential roles in regulating gene expression. In our analysis, we found various classes of ncRNAs were methylated both before and during copper-induced oxidative stress response. Among them, lncRNAs had the highest number of transcripts that contained stress-induced and stress-depleted m^6^A. We were curious whether we would see a positive change in transcript abundance in the lncRNA transcripts with the stress-enriched m^6^As. On the contrary, we saw a negative change in transcript abundance for the lncRNAs, which suggests that the m^6^A-dependent regulation of lncRNA may differ from that of ncRNA compared to mRNA. This interesting finding will be explored in future research.

## 4. Materials and Methods

### 4.1. Stress Treatment for RNA-Seq, m^6^A-IP-Seq, and Polysomal RNA-Seq

Surface-sterilized *Arabidopsis thaliana* seeds from Col-0 accession were plated on ½ Murashige and Skoog (MS) agar medium, vernalized at 4 °C for 48 h under dark conditions, then were transferred to a 22 °C growth chamber where the plants were grown under a 12/12 h light and dark cycle for three weeks. For inducing oxidative stress, seedlings were then sprayed with 100 μM CuSO_4_ while seedlings without treatment served as controls. After 24 h of treatment, both control and copper-treated seedlings were harvested and stored at −80 °C until RNA extraction.

### 4.2. m^6^A RNA Immunoprecipitation and Sequencing (m^6^A-IP-Seq)

The immunoprecipitation of m^6^A-containing RNA fragments was conducted as described previously [24]. Briefly, total RNA was extracted from the copper-treated and control samples using TRIzol and the quality of RNA was assessed using 1.2% agarose gel. Approximately 120 μg of total RNA was then used for poly(A) RNA isolation using Arraystar Seq-StarTM poly(A) mRNA Isolation Kit separately for three biological replicates. Purified polyadenylated RNA was chemically fragmented using 10 mM Zn^2+^ at 94 °C for 5 min in 10 mM Tris-HCl (pH 7.0) buffer. After analyzing the size of the fragments using an Agilent 2100 Bioanalyzer, an aliquot of the fragmented polyA^+^ RNA was immunoprecipitated using 2 μg anti-m^6^A rabbit polyclonal antibody (Synaptic Systems) in 500 μL of the IP buffer (10 mM Tris-HCl at pH7.4, 150 mM NaCl, 0.1% NP-40) at 4 °C for 2 h. 20 μL Dynabeads^TM^ M-280 Sheep Anti-Rabbit IgG was prepared and incubated with the IP mixture for 2 h at 4 °C and washed 3 times with IP buffer and 2 times with wash buffer (10 mM Tris-HCl at pH7.4, 50 mM NaCl, 0.1% IGEPAL CA-630). The resulting immunoprecipitated m^6^A-containing RNA fragments were eluted in a 10 mM Tris-HCl at pH7.4 containing 1 mM EDTA, 0.05% SDS, and 40U proteinase K at 50 °C for 30 min. The RNA fragments were purified using the phenol–chloroform extraction protocol and precipitated with ethanol. The resulting m^6^A-containing mRNA fragments were used to construct m^6^A-IP-seq library using a KAPA stranded mRNA-seq Kit (Illumina^®^ platform) and sequenced on the Illumina HiSeq 4000 platform. In parallel, aliquots of the fragmented polyA^+^ RNA without the immunoprecipitation were used to generate three biological replicates of background polyA^+^ RNA-seq libraries for both control and stress-treated samples.

### 4.3. Polysomal RNA-Sequencing

The polysomes were isolated from three-week-old Arabidopsis seedlings of both control or copper-treated for 24 h seedlings as described previously [37]. For treated and untreated samples, two biological replicates were ground into a fine powder using liquid nitrogen and used for polysome extraction. Approximately 500 mg of powdered tissue was thoroughly resuspended in 1250 μL polysome extraction buffer (200 mM Tris (pH 9.0), 200 mM KCl, 26 mM MgCl_2_, 25 mM EGTA, 100 μM 2-mercaptoethanol, 50 μg/mL cycloheximide, 50 μg/mL chloramphenicol, 1% Triton X-100, 1% Brij-35, 1% Tween-40, 1% IGEPAL CA-630, 2% polyoxyethylene-10-tridecyl ether, 1% deoxycholic acid) and incubated for 10 min on ice with occasional mixing. The polysome was separated using centrifugation at 14,000 rpm for 2 min at 4 °C. The resulting supernatant fraction was layered on sucrose density gradients (20–60% sucrose, *w*/*v*) and centrifuged for 120 min at 40,000 rpm (275,000× *g*) in a Beckman OPTIMA LE-80 centrifuge. Using UA-5 detector and a Gradient Fractionator (model 640, ISCO), optical densities of the gradient fractions were measured at 254 nm. For polysome RNA, sucrose fractions with more than two ribosomes were pooled and extracted using TRIzol reagent. Approximately 8 μg polysome associated RNA per replicate was subjected to polyA^+^ selection followed by RNA-seq library preparation using KAPA Stranded mRNA-seq Kit (Illumina^®^ platform). After validating the library quality using Agilent 2100 Bioanalyzer, the polysomal sequencing library was sequenced using Illumina HiSeq 4000 platform.

### 4.4. Bioinformatics Analysis of RNA-Seq, m^6^A-IP-Seq, and Polysome-Seq Data

#### 4.4.1. Read Processing

Sequencing files generated from HiSeq 4000 platform in raw fastq format were quality checked using the bioinformatic package FastQC [38]. Using Trimmomatic-0.36, the Illumina sequencing adapters were trimmed from raw reads using the tool Trimmomatic-0.36 and the provided Truseq3-SE adapter sequences [39]. Trimmed fastq sequences were mapped to Arabidopsis genome assembly TAIR10 downloaded from the ENSEMBL database using STAR aligner to generate BAM files using parameters: --outSAMattributes All and -outSAMtype BAM [40].

#### 4.4.2. Differential RNA Abundance

To calculate the RNA abundance, the count feature of python package HTSeq [41] was used to assign the number raw reads in the BAM files to gene IDs annotated in the TAIR10 reference annotation file obtained from ENSEMBL repository in a strand-specific manner. These raw counts were then used as input for an R statistical package DESeq2 [42] using its default settings, which uses normalization factors that incorporate library depth and gene wide dispersions for normalization of the reads and determines the change in expression in the form of fold change (log converted) between control and stress treatments and respective *p*-values of the difference. Transcripts that showed adjusted *p*-value of <0.05 were called significant.

#### 4.4.3. m^6^A Peak Calling

As described previously, we used peak calling software MACS2 to call m^6^A peaks [43]. Briefly, peaks were called using the alignment bam files of the m^6^A-IP-seq and the background mRNA-seq as input with parameters as described previously [34]. Briefly, in order to preserve the strand information, the reads in the bam files were split using SAMtools [44] based on the strand they aligned to. Peaks were then called separately for the plus and minus strands for each condition and each replicate using parameters –nomodel. The input polyA RNA that was used as background for STAR was also split based on plus and minus strand. Peaks were filtered by *p*-values < 0.05. The output was a NarrowPeak file that contained the loci of peaks identified by MACS2.

#### 4.4.4. Motif Searches

Sequence motifs enriched in the m^6^A peak regions in the NarrowPeak files were searched using the de novo motif search algorithm HOMER [45]. In simple words, HOMER looks for sequence features that are enriched in a list of provided sequences against randomized background of nucleotides reads with matched GC% content along with the possible false positive ratio. In this case, the sequences provided to HOMER were extracted by comparing the peak loci in the NarrowPeak files to the annotated reference transcript sequence.

#### 4.4.5. m^6^A Frequency

For the transcripts grouped by whether the m^6^A it contained was enriched, depleted, or remain unchanged by copper-induced oxidative stress, the frequency of m^6^A methylation for each nucleotide position near the start or stop codon was calculated using the following formula:frequency = (number of unique m^6^A peak regions that overlapped with that nucleotide position/the total number of transcripts that were m^6^A methylated)

The frequency value for each category of transcripts was then normalized to the maximum frequency value within the 200 nt region flanking the start or stop codon to visualize the trend of m^6^A in these boundary regions in each category of transcripts.

#### 4.4.6. Polysome Occupancy

The polysome occupancy of each transcript was determined by first normalizing the raw read count of each transcript within each replicate to the sequencing depth of the library and calculating the read per million (RPM) value. The polysome occupancy value for each transcript was then calculated as:polysome occupancy = log_2_ (RPM from polysome sequencing/RPM in the RNA-seq).

In order to determine the effect of copper stress on polysome occupancy of RNA, we calculated the change in occupancy value for each transcript with and without stress using the formula:Polysome occupancy change = Polysome Occupancy_copper_ − Polysome Occupancy_control_

The difference is the log transformed polysome occupancy values where a negative value represents a decrease in polysome occupancy while a positive value means an increase in occupancy upon oxidative stress.

#### 4.4.7. Statistical Analysis

For differential abundance analysis, we used DeSeq2, which conducts a Wald test and Bonferroni correction to give an adjusted *p*-value. For comparing relative difference in mRNA abundance between transcripts that had stress-induced m^6^A, control-enriched m^6^A, and shared m^6^A-containing transcript groups, we used Wilcoxon rank sum tests and the results were considered statistically significant for *p*-values < 0.05. For designating anything statistically significant, we used padj < 0.05 values. A similar method was used to test the statistical significance of the differences in ribosome occupancy values. For comparing the relative abundance of non-coding RNA groups, we used two sample *t*-tests and considered the results significant for *p*-values < 0.05.

## Figures and Tables

**Figure 1 ncrna-10-00008-f001:**
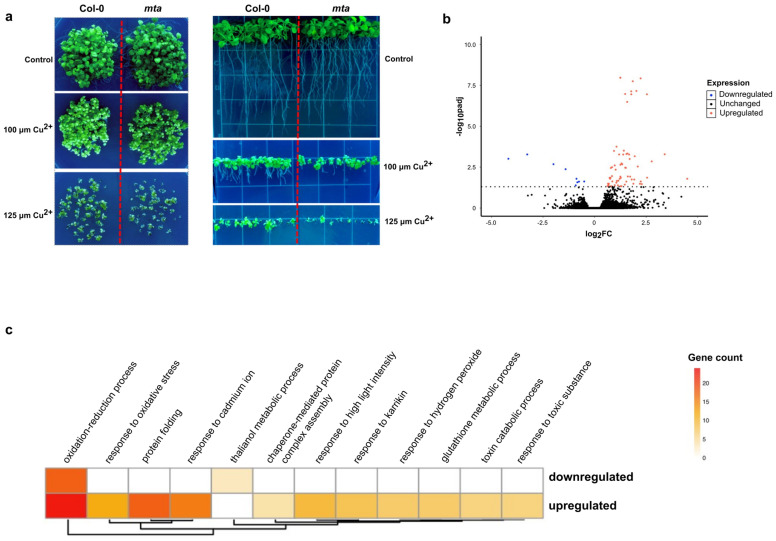
m^6^A is involved in plant copper-induced oxidative stress response. (**a**) Comparison of growth phenotypes between Col-0 and *mta* mutant seedlings under copper stress showing overall germination and survival when grown horizontally (**left**) and vertically (**right**) in a ½ strength MS agar supplemented with 100 μM Cu^2+^ and 125 μM Cu^2+^ in the media. Overall survival and root and shoot growth are all more severely affected in the *mta* mutant compared to Col-0 with increased strength of the stress. (**b**) Volcano plot showing the transcriptome-wide change in transcripts upon copper stress for 24 h. The red and pink dots represent transcripts that met our threshold of significant decrease and increase (*p*-value < 0.05). (**c**) Heatmap showing the number of genes significantly associated with various functional terms upon copper stress via GO analysis.

**Figure 2 ncrna-10-00008-f002:**
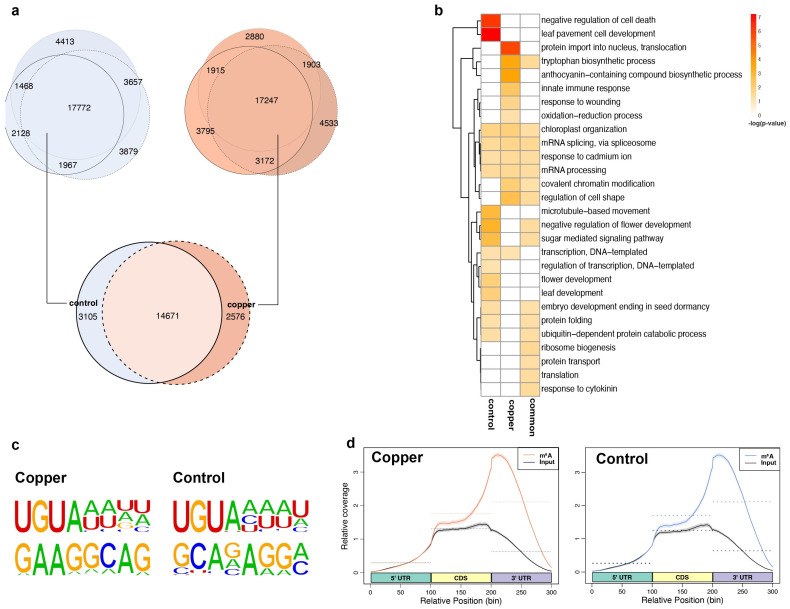
mRNA m^6^A dynamics in response to copper-induced oxidative stress. (**a**) Venn diagrams showing overlap between the m^6^A peaks identified in the biological replicates within control (**top left**) and copper (**top right**) treatments. The overlapping peaks (or high-confidence peaks) are then used to compare the m^6^A peaks detected between control and copper conditions (**bottom**). (**b**) Gene ontology for transcripts containing control-enriched, copper-enriched, or common m^6^A peaks between treatments. (**c**) Two of the most significantly enriched motifs found in the m^6^A peak regions containing a canonical A site in both the copper and control conditions. (**d**). Distribution of the m^6^A-IP-seq reads along the binned regions (5′ UTR, CDS, and 3′ UTR) of Arabidopsis transcripts, as specified.

**Figure 3 ncrna-10-00008-f003:**
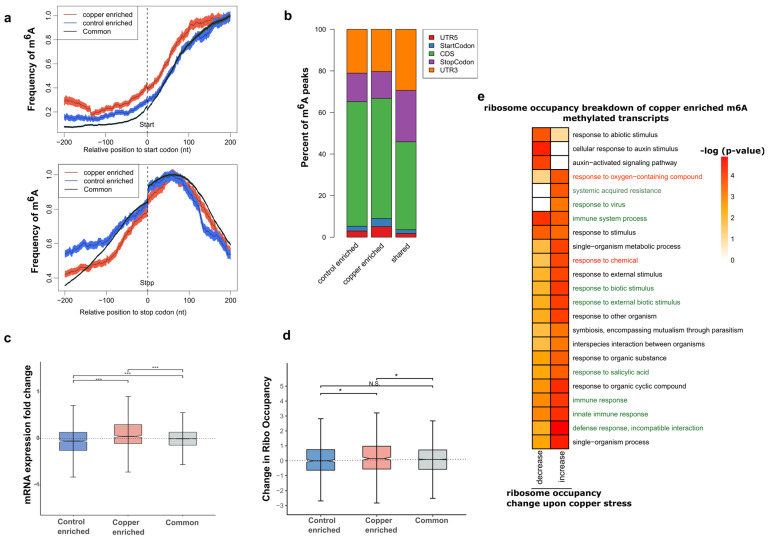
Copper-induced oxidative stress-specific m^6^A sites affect the abundance and translation of transcripts encoding proteins relevant to stress response. (**a**) Per-nucleotide position frequency of methylation in the 200 nt region around the start codon (**left**) and stop codon (**right**). (**b**) Percent breakdown of m^6^A peak regions in each category based on the region of transcript they overlap. (**c**) Change in RNA abundance, measured as log_2_ fold change between copper stress and control for transcripts that contain either copper-enriched m^6^A, control-enriched m^6^A, or common m^6^A peaks, are plotted in a box and whisker plot. All differences are significant (*** *p*-value < 0.001) based on Wilcoxon rank sum tests. (**d**) Box and whisker plot of change in ribosome occupancy between control and copper stress conditions in the three categories of m^6^A methylated transcripts based on whether the methylation is enriched under stress only, enriched under control only, or found in both conditions. Significance was measured using Wilcoxon rank sum tests and * denotes *p*-value < 0.01 for two of the three comparisons as indicated, while N.S. describes the comparison that is not significantly different. (**e**) Heatmap showing the functional terms enriched by GO analysis for transcripts that contain stress-enriched m^6^A based on whether they show positive or negative change (also known as increase or decrease) in polysome association. The green highlighted terms are related to defense mechanisms and the red highlighted terms are related to oxidative stress response.

**Figure 4 ncrna-10-00008-f004:**
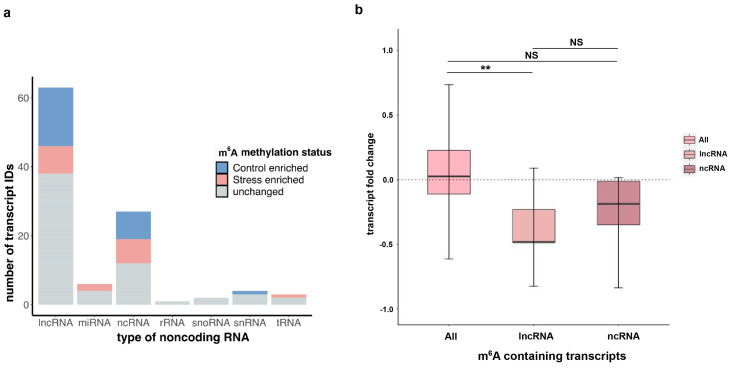
Long non-coding RNA (lncRNA) m^6^A dynamics in response to copper-induced oxidative stress. (**a**) Stacked bar graph showing the total number of unique non-coding RNAs of various known categories or labelled as ncRNA if category is not clear in the annotation. Breakdown of ncRNA based on number of transcripts containing stress-enriched m^6^A (red bar), control-enriched m^6^A (blue bar), or m^6^A that is common or unchanged between conditions (grey bar). (**b**) For ncRNA and lncRNA transcripts containing stress-induced m^6^A, box and whisker plot showing change in mRNA levels after copper stress treatment. Significance was measured using Wilcoxon rank sum tests and ** denotes *p*-value < 0.001 for the comparison as indicated, while N.S. describes the two comparisons that are not significantly different.

**Figure 5 ncrna-10-00008-f005:**
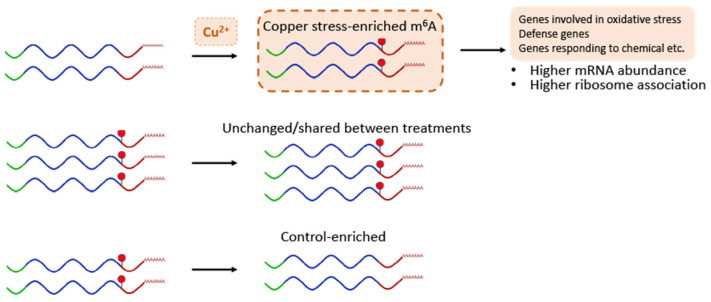
A model for m^6^A function in Arabidopsis copper-induced oxidative stress response. The model depicts the identified m^6^A dynamics for transcripts gaining and losing this modification during copper-induced oxidative stress response. Our data also suggest that the gain and loss of m^6^A has effects on the stability and ribosomal association of these transcripts during this stress response, allowing plants to properly respond.

## Data Availability

The sequencing data presented in this manuscript can be found in NCBI SRA under the accession number: PRJNA1061764.

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
