# Peer review of "RNA N6-Methyladenosine Affects Copper-Induced Oxidative Stress Response in Arabidopsis thaliana"

_ncrna, 2024, doi:10.3390/ncrna10010008_

Round 1

Reviewer 1 Report

Comments and Suggestions for Authors

Dear authors,

Find my comments in the enclosed file.

Author Response

What is mta? Please clarify it in the abstract section.

Ans: corrected.

Keywords should be different from those found in the title, please change the keywords that are the same from the article title.

Ans: deleted the keywords present in the title and added the following new keywords: epitranscriptome, transcript stability, long noncoding RNAs

lack of reference

Ans: References are added.

Remove "species" from here since it is referring to ROS and the "s" from ROS is related to "species".

Ans: corrected.

All genes involved in a same function were downregulated or upregulated? For example, all transcripts from genes involved in protein folding were downregulated?

Ans: For this analysis, only genes that were upregulated or downregulated based on differential expression were evaluated for Gene Ontology analysis. Of those that are regulated (which is already a small subset of the total number of genes involved in a given process), a clear trend was observed for almost all of these transcripts for the functions shown. It is important to note that by the nature of GO analysis, certain functions may not show up as significant due to low sample size of genes.

In my point of view, the place of the figures should be changes to appear after its description in the results section.

Ans: corrected.

It should be copper condition.

Ans: Corrected.

Take a look at the description of each subfigure. Subfigure B show the function of the trasncripts, subfigure C show the motifs in the m6A and D show the peaks of the m6A. Also, change the colors of the input and m6A in subfigure D because they are too similar and hard to separate specifically in the legend.

Ans: Corrected.

Look that sometimes appears cold stress, please check it along the manuscript.

Ans: Corrected.

It should be figure 3b. Please check.

Ans: Corrected.

the lack of articulation between the results found in this study and the result in the literature means that this section is not a discussion but rather a description of the results. Furthermore, you make important considerations when describing the results. You have a lot of important results and as stated along the text, you have results published previously. Please, rewrite the discussion section to enrich your manuscript.

Ans: There are almost no published reports available on the m6A analysis specifically under oxidative stress conditions. Wherever relevant, the impact of altered m6A on transcript stability and translational enhancements were incorporated in the discussion.

temperature?

Ans: Corrected.

Standardize your units along the text.

Ans: Corrected.

How were they considered upregulated or downregulated? The log2fold change different from "zero"? Higher and lower than 1?

Ans: Anything that was higher or lower than log2fold change of 0 is called down or upregulated as long as the difference is statistically significant (p <0.05)

Reviewer 2 Report

Comments and Suggestions for Authors

This study by Sharma et al. investigates the changes in m6A methylation profiles across the transcriptome during oxidative stress response in Arabidopsis. By subjecting seedlings to excess copper ions, an inducer of oxidative stress, the researchers performed m6A RNA immunoprecipitation sequencing (me-RIP-seq) to examine alterations in m6A status of transcripts. Additionally, they assessed changes in polysome binding of transcripts with responsive m6A enrichment. The results provided intriguing insights into the differential methylation patterns and their association with changes in transcript abundance. The findings indicate that while there was a modest change in the overall RNA expression profile in response to oxidative stress, genes associated with oxidative stress response showed significant alterations. Moreover, a substantial number of transcripts exhibited changes in m6A status, suggesting a role for epitranscriptomic regulation in oxidative stress response. Additionally, an intriguing observation was made regarding the differential regulation of coding and non-coding RNAs in response to oxidative stress, suggesting distinct mechanisms underlying their epitranscriptomic regulation.  The study's findings open new avenues for future research, prompting further investigations into the molecular mechanisms and specific m6A reader proteins that mediate the epitranscriptomic regulation of oxidative stress response in plants. Understanding this intricate regulatory network holds potential for advancing stress tolerance strategies in crops, with broad implications for agricultural resilience in changing environmental conditions.

I have a few concerns that can be addressed:

1. Line 92, full form of MTA should be mentioned.

2. There is no mention of 'defense response to bacterium' in Figure 1c as mentioned in Line 135. Reference should be added if referred to a previous study.

3. Figure 2b and 2c have been mislabelled.

4. Figure 2b (which is mislabelled 2c) missing labeled axis.

5. Figure 3a and 3b have been mislabelled.

Author Response

I have a few concerns that can be addressed:

  1. Line 92, full form of MTA should be mentioned.

Ans: Suggestion incorporated.

  1. There is no mention of 'defense response to bacterium' in Figure 1c as mentioned in Line 135. Reference should be added if referred to a previous study.

Ans:  It should be “innate immune response” and this term is replaced the ‘defense response’ term now.

  1. Figure 2b and 2c have been mislabeled.

Ans: Corrected.

  1. Figure 2b (which is mislabeled 2c) missing labeled axis.

Ans: Corrected.

  1. Figure 3a and 3b have been mislabeled.

Ans: Corrected.

Reviewer 3 Report

Comments and Suggestions for Authors

In the manuscript by Sharma et al., the authors propose that m6A plays a significant role during the oxidative stress response in Arabidopsis thaliana. The assessment of germination and seedling development of the mta mutant subjected to heavy copper treatment revealed the relevance of the m6A epitranscriptome in plant oxidative stress tolerance.

The data presented in the manuscript is interesting. The manuscript is well written. Some further experiments need to be done in order to explore the gene regulation associated with the event.

I have a few major/minor suggestions for polishing the manuscript.

Major Comments:

1.     In figure 1C, the authors claim that “transcripts (e.g, GSTU2, GSTU4 etc.) associated with highly relevant biological processes were significantly altered in their abundance levels.” The authors should check the relative steady state levels of a few of these targets in their data using RT-qPCR. It will further strengthen their data biologically.

2.     Did the authors find any commonality between the RNA-seq data in presence of copper induced stress and m6A-IP-seq data upon copper induced oxidative stress. E.g., Did they find any upregulated transcripts in RNA-seq data in presence of copper induced stress that also undergoes m6A methylation event.  The authors should comment on this.  They can also include a Venn diagram in order to explain this.

3.     In Figure 3C, the authors should comment on why transcripts that contain copper stress enriched m6A have increased upregulation.  Just like Comment 1, the authors should pick up a few targets and check the relative steady state levels using RT-qPCR. The authors should also do an mRNA stability analysis or ChIP or check the reason behind this upregulation.

4.     In figure 4b, the authors have pointed out that ncRNA transcripts showed a negative transcript abundance upon stress contrary to mRNA transcripts.  This looks like a wonderful observation.  The authors should try to find some sense/antisense gene pairs from their datasets and revalidate the data using RT-qPCR.  This can point out a unique mechanism of how an m6A switch has a crosstalk with sense-antisense mediated regulation.  This will also strengthen their data on increased polysome association with the m6A transcripts.

5.     The authors should also include a graphical representation of the work for the broader audience.

Minor Comments

1.     In the Introduction Section, Page 2, Line 64-68, the authors have cited a few examples that explore the role of m6A in various stress responses associated with the plants.  The authors should cite some more examples that depict the cross-talk between m6A and stress responses.

2.     The authors should maintain uniformity in the figures.  Figure 4b and 3C denote the RNA fold change. The authors have denoted the level of significance by ** in Figure 3C and by a p-value number in Figure 4b. The authors should adopt a single procedure in both the figures.

3.     There are already some previous reports on the enrichment of m6A peaks on the 5’UTR of the transcripts.  Discuss that in the discussion section.  This will strengthen their claims.

4.     The authors should include a separate statistical section in the Materials and Methods Section.

Author Response

Major Comments:

  1. In figure 1C, the authors claim that “transcripts (e.g, GSTU2, GSTU4 etc.) associated with highly relevant biological processes were significantly altered in their abundance levels.” The authors should check the relative steady state levels of a few of these targets in their data using RT-qPCR. It will further strengthen their data biologically.

Ans: Thank you for this suggestion.  We believe that the use of three biological replicates in our sequencing analyses is more robust that follow-up qPCR analyese. Thus, we have included RNA-seq browser views showing the increased mRNA levels of example transcripts across 3 independent biological replicates in both control and copper stress treated conditions and the below figure is added for reviewers’ reference and to the manuscript as Supplemental Figure 2. We hope that this increases the confidence in our data.

  1. Did the authors find any commonality between the RNA-seq data in presence of copper induced stress and m6A-IP-seq data upon copper induced oxidative stress. E.g., Did they find any upregulated transcripts in RNA-seq data in presence of copper induced stress that also undergoes m6A methylation event.  The authors should comment on this.  They can also include a Venn diagram in order to explain this.

Ans: Approximately ~18% (12 out 69 transcripts) of significantly upregulated transcripts in mRNA seq also found to possess stress-induced m6A upon copper stress compared to none of the downregulated mRNA transcripts (9) that showed induced m6A upon copper stress.  Furthermore, among transcripts containing copper-enriched m6A, 27 showed > 50% decrease, whereas 139 showed > 50% increase in mRNA levels after copper stress treatment. This inconsistency may warrant further investigation and we will take up this in our follow-up experiments. We have added this caution in the revision now.

A few examples of browser views are presented in supplemental figure 1, showing such correlations between the RNA-seq and m6A-IP-seq in presence of copper induced oxidative stress.

  1. In Figure 3C, the authors should comment on why transcripts that contain copper stress enriched m6A have increased upregulation.  Just like Comment 1, the authors should pick up a few targets and check the relative steady state levels using RT-qPCR. The authors should also do an mRNA stability analysis or ChIP or check the reason behind this upregulation. 

Ans: We thank the reviewer for the suggestion. While the exact mechanism of why copper stress enriched m6A may lead to increased upregulation of certain transcripts has not been fully established, it is possible that m6A reader proteins that bind to the m6A modification are involved in this pathway. Recent studies have suggested YTH-domain containing proteins ECT2/3/4 are involved in transcript stabilization of stress related transcripts under ABA stress (Song et al., Genome Biology, 24, 103, 2023). Future experiments involving the mutants of these reader proteins under copper induced oxidative stress may be an interesting avenue of addressing this question. This note is added in the discussion section now.

  1. In figure 4b, the authors have pointed out that ncRNA transcripts showed a negative transcript abundance upon stress contrary to mRNA transcripts.  This looks like a wonderful observation.  The authors should try to find some sense/antisense gene pairs from their datasets and revalidate the data using RT-qPCR.  This can point out a unique mechanism of how an m6A switch has a crosstalk with sense-antisense mediated regulation.  This will also strengthen their data on increased polysome association with the m6A transcripts.

Ans: We have not analyzed sense/antisense gene pairs using our current dataset, we would like to address this by critically analyzing the large-scale datasets in future.

  1. The authors should also include a graphical representation of the work for the broader audience.

Ans: We have added a new figure 5 now in the discussion.

Minor Comments

  1. In the Introduction Section, Page 2, Line 64-68, the authors have cited a few examples that explore the role of m6A in various stress responses associated with the plants.  The authors should cite some more examples that depict the cross-talk between m6A and stress responses. 

Ans: We have added almost 15 relevant references while we are introducing the m6A and then added 3 pioneering references on stress responses. Hope this should be sufficient for this manuscript.

  1. The authors should maintain uniformity in the figures.  Figure 4b and 3C denote the RNA fold change. The authors have denoted the level of significance by ** in Figure 3C and by a p-value number in Figure 4b. The authors should adopt a single procedure in both the figures.

Ans: Figure 4 has been updated for consistency and the level of significance is indicated by ** in both figures.

  1. There are already some previous reports on the enrichment of m6A peaks on the 5’UTR of the transcripts.  Discuss that in the discussion section.  This will strengthen their claims.

Ans: Without stress, we are only finding enrichment of m6A peaks in the 3’UTR and this seems to be the case with almost all published Arabidopsis m6A analysis reports. Our statement is restricted to the stress-responsive m6A peak localization (but not the overall m6A peak localization distribution along the transcripts), i.e., a small fraction of the stress enriched or depleted m6A peaks are slightly more likely to be found on the 5’ UTR, start codon, and CDS regions. Unable to appreciate how our analysis will strengthen the enrichment of m6A peaks on the 5’UTR of the transcripts.

  1. The authors should include a separate statistical section in the Materials and Methods Section.

Ans: Below description is added now in the Materials and Methods Section.

For differential expression analysis, we used DeSeq2 which conducts Wald test and Bonferroni correction to give an adjusted p-value. For comparing relative difference in mRNA abundance between transcripts that had stress induced m6A, control enriched m6A and shared m6A containing transcript groups, we used Wilcoxon rank sum test and considered statistically significant if p-value < 0.05. For designating anything statistically significant, we used padj<0.05. Similar method was used for comparing differences in ribosome occupancy values.  For comparing the relative abundance of non-coding RNA groups, we used two sample t-test and considered significant if p-value <0.05.

Round 2

Reviewer 3 Report

Comments and Suggestions for Authors

In the manuscript by Sharma and Govindan et al., the authors propose that m6A plays a significant role during the oxidative stress response in Arabidopsis thaliana. The assessment of germination and seedling development of the mta mutant subjected to heavy copper treatment revealed the relevance of the m6A epitranscriptome in plant oxidative stress tolerance.

The authors have addressed all the previous comments. Thus, the manuscript can be accepted in its present form.